# Preparation of Hydrogels Based on Modified Pectins by Tuning Their Properties for Anti-Glioma Therapy

**DOI:** 10.3390/ijms24010630

**Published:** 2022-12-30

**Authors:** Andrei Belousov, Aleksandra Patlay, Vladimir Silant’ev, Valeri V. Kovalev, Vadim Kumeiko

**Affiliations:** 1Institute of Life Sciences and Biomedicine, Far Eastern Federal University, Vladivostok 690922, Russia; 2Laboratory of Electrochemical Processes, Institute of Chemistry, FEB RAS, Vladivostok 690022, Russia; 3A.V. Zhirmunsky National Scientific Center of Marine Biology, FEB RAS, Vladivostok 690041, Russia

**Keywords:** natural polymers, polysaccharides, extracellular matrix, glioblastoma, viscoelastic properties

## Abstract

The extracellular matrix (ECM) of the central nervous system (CNS), characterized by low stiffness and predominance of carbohydrates on protein components, mediates limited cell proliferation and migration. Pectins are polysaccharides derived from plants and could be very promising for a tunable hydrogel design that mimics the neural ECM. Aiming to regulate gel structure and viscoelastic properties, we elaborated 10 variants of pectin-based hydrogels via tuning the concentration of the polymer and the number of free carboxyl groups expressed in the degree of esterification (DE). Viscoelastic properties of hydrogels varied in the range of 3 to 900 Pa for G′ and were chosen as the first criteria for the selection of variants suitable for CNS remodeling. For extended reciprocal characterization, two pairs of hydrogels were taken to test pectins with opposite DEs close to 0% and 50%, respectively, but with a similar rheology exceeding 100 Pa (G′), which was achieved by adjusting the concentration of pectin. Hydrogel swelling properties and in vitro stability, together with structure characterization using SEM and FTIR spectroscopy, displayed some differences that may sense for biomedical application. Bioassays on C6 and U87MG glioblastoma cultures testified the potential prospects of the anti-glioma activity of hydrogels developed by decreasing cell proliferation and modulating migration but supporting the high viability of neural cells.

## 1. Introduction

Biopolymers are widely used in biotechnology and regenerative medicine in the form of hydrogels, which are crosslinked polymer mesh structures capable of absorbing large amounts of water. Due to this feature, as well as their elasticity and porosity, hydrogels mimic the native microenvironment of living tissues and are used as artificial analogues of the ECM [1,2]. Natural hydrogels, unlike most synthetic ones, are biodegradable, often biocompatible and non-toxic, and can regulate cellular behavior because they contain specific ligands to receptors on the cell surface.

The traditional approach to cancer treatment focuses only on tumor cells, ignoring their non-cellular environment, in particular their ECM. However, it was proven that the key role in the development and progression of a tumor is played not by the tumor cells themselves, but by their microenvironment. Understanding this role is critical to predicting the fate and behavior of the remaining cancer cells and the dynamics of recovery [3].

In regenerative medicine and tissue engineering, the interaction of cells with biomaterials is important for the proper functioning of tissues during repair and regeneration processes. Biomaterials used for implantation must mimic the mechanical characteristics of the tissue they are intended to replace [4]. The normal brain has a storage modulus in the range of 100–1000 Pa [5], while glial tumors have more than 1000 Pa, depending on the measurement method [6,7]. At the same time, artificial substrates support normal and cancer cells of the nervous system even at much higher rigidity. Glioma cells prefer more rigid substrates, with a Young’s modulus greater than 10,000 Pa [8]. An increase of brain stiffness is associated with an expansion of ECM in the tissue up to 48% in grade III and IV astrocytomas [9], while this parameter in the normal brain is about 20% [10]. For effective anti-glioma therapy, materials with a storage modulus less than 1000 Pa can be used, similar to a normal brain, and that are capable of slowing down the growth and development of a neoplasm. Carbohydrate polymers, including polysaccharides and glycosaminoglycans, predominate in the ECM of the brain over highly adhesive protein components. This mediates the low stiffness of a normal ECM in the adult brain, supporting limited cell migration and proliferation.

The acidic glycopolymers of the human brain can be mimicked by plant-derived polysaccharides, usually composed of uronic acid derivatives. Pectin is a linear anionic plant polysaccharide that has the ability to form hydrogel upon heating or mixing with divalent cations, consisting predominantly of polygalacturonic acid residues partially or wholly methoxylated. The number of esterified carboxyl groups for every 100 free ones is called the degree of esterification (DE). High-esterified pectins (DE over 50%) are common in nature, but low-esterified pectins (DE less than 50%) can be obtained by chemical de-esterification [11].

The prospects for the use of pectin hydrogels in biomedicine are associated with their biocompatibility, biodegradability, low toxicity, the possibility of their physical performance, the ability to immobilize cells, proteins, drugs, or growth factors [12]. Pectin-based hydrogels are able to imitate native brain ECM, the main polysaccharide component of which is hyaluronic acid, while pectin hydrogels regulate the dynamics of brain tumor cells [13]. In the postoperative therapy of brain tumors, bioactive pectin-based implants can be used for an inhibitory effect on residual tumor cells in the resection cavity, followed by replacement with a material that supports neurite growth [14].

In this work, hydrogels based on pectins with different degrees of esterification and similar mechanical properties were prepared by ionic gelation. The hydrogels were characterized by rheometric analysis, Fourier-transform infrared spectroscopy (FTIR), and scanning electron microscopy (SEM). Viscoelastic properties of hydrogels were chosen as the first criteria for the selection of variants suitable for brain ECM remodeling. By tuning the concentration of pectin and the content of free carboxyl groups, hydrogels, characterized by appropriate physicochemical properties and maintaining cell viability, were developed and proven to be a matrix that slows down cell proliferation and migration, which is of interest for promising anti-glioma therapy.

## 2. Results

In the present study, a new implantable hydrogel was developed based on standardized pectin samples. The addition of Ca^2+^ to them as a gelation initiator made it possible to form a stable polymer network from pectin. Preliminary tests (data not shown) made it possible to choose the best Ca^2+^ concentrations capable of inducing a sol-gel transition with the formation of homogeneous stable gels under physiological conditions. Various hydrogel formulations were characterized physicochemically, and preliminary biological characterization was performed to evaluate the systems’ ability to slow metabolism and modulate cancer cell behavior.

### 2.1. Pectin Substrate

Pectin samples with degrees of esterification of 1.2% and 52.0% were obtained by alkaline de-esterification.

Physicochemical parameters of the pectin samples obtained are shown in Table 1.

It was established that pectin samples that undergo the de-esterification procedure are characterized by a similar content of anhydrogalacturonic acid, ranging from 72.0% to 73.7% by weight of the sample, which was slightly lower than the content in the original pectin. Differences in the content of anhydrogalacturonic acid can be explained by the partial degradation of uronic acids and the removal of side chains during the alkaline esterification process. Additionally, neutral sugars were washed out by rinsing with ethanol. During de-esterification, there was a decrease in intrinsic viscosity from 915 mL/g of the original pectin to 570 mL/g for the sample with DE 50% and 408 mL/g for the de-esterified pectin, which indicates a decrease in the molecular weight of the samples during processing.

### 2.2. FTIR

FTIR spectra of pectin 0 powder and its 1.25% gel are shown in Figure 1. Among the absorption bands general to both spectra, they are in the ranges 3900–3500 cm^−1^ (stretching vibration of ν(OH)) and intramolecular H-bonds) and 3000–2900 cm^–1^ (C-H stretching of ν(CHx)). Peaks at 1547 and 1530 cm^−1^, respectively, were attributed to ν_as_(COO-) stretching vibrations of carboxylic acid salts. The “fingerprint” region 1100–600 consists of bands related typically to polysaccharide C–O stretching vibrations of ν(C-O) and C-O-C bridges. The area of carboxyl groups consists of several intense peaks: 1731 and 1737 cm^−1^, respectively, corresponding to the ν(C=O) ester stretching vibration esterified carboxylic group. The shoulder at the peak at 1642 cm^−1^ from the gel spectrum belongs to the methyl esterified carboxyl group antisymmetric stretching vibration of ionic (protonated) carboxyl groups ν_as_ (COO-), which determine their participation in the ionic crosslinking process during gel formation. Moreover, the peak at the 1420–1400 region may be attributed to ν_s_ (COO-) symmetric stretching vibrations and shifts to low frequencies in the gel spectrum.

The FTIR spectra of pectin 50 powder and its 3% gel are shown in Figure 2. The range of stretching vibration of ν(OH) and intramolecular H-bonds of the pectin 50 is wider and is located in a higher frequency region compared to the spectrum of its gel. This difference compared to the previous pectin 0 sample was repeated in subsequent measurements and probably lies in the method of preparation of the pectin itself. Other general ranges of vibrations in the spectra of pectin 50 and the gel obtained from this polysaccharide are correlated: 3000–2900 cm^−1^ is the C–H stretching of ν(CH_x_); peaks at 1600 and 1530 cm^−1^, respectively, have a large difference compared to the previous pair spectra from Figure 1, and this is attributed to ν_as_(COO-) stretching vibrations of carboxylic acid salts. The “fingerprint” region 1100–600 consists of the same bands related to the C–O stretching vibrations of ν(C-O), and C-O-C bridges are more intense on the spectrum of pectin powder. The vibrations of ν(C=O) and the ester stretching vibration esterified carboxylic group are located at 1731 and 1736 cm^−1^, respectively. The gel spectrum has a similar shoulder at 1642 cm^−1^ that belongs to protonated carboxyl groups. The peak of the 1420–1400 region that was attributed to ν_s_(COO^−^) symmetric stretching vibrations appeared just in the gel spectrum.

### 2.3. Rheological Properties

The viscoelastic properties of modified pectin hydrogels were studied by rheological analysis by obtaining data on the dependence of the storage modulus (G′) that measures the stored energy, representing the elastic portion (unlike the loss modulus (G″) that measures the energy dissipated as heat, representing the viscous portion), on the concentration of the polysaccharide powder *w*/*v* and DE. The analysis showed an exponential dependence of G′ on the concentration for both types of samples, P0 and P50 (Figure 3a). G′ was increased from 3.8 ± 0.3 Pa to 873.4 ± 46.8 Pa for P0 samples (concentration *w*/*v* from 0.25% to 1.65%) (n = 3); in comparison, P50 G′ increased from 12.2 ± 1.8 Pa to 574.0 ± 49.4 Pa (concentration *w*/*v* from 1.5% to 5.0%) (n = 3). This means that hydrogels from pectin with a DE close to 0% are generally tougher than hydrogels from pectin with a DE close to 50%.

The pairs of samples P0-1.25% and P50-3%, P0-1.5% and P50-4.5%, with similar values of G′, about 125.7 ± 17.5 Pa and 396.3 ± 47.9 Pa, respectively (n = 3), and pair P0-1.5% and P50-1.5% (n = 3) with different G′ values, despite the same concentration, were chosen for further analysis.

The dependence of G″ on the concentration of pectin is similar to the G′. G″ increased from 1.1 ± 0.2 Pa to 82.1 ± 11.6 Pa (n = 3) for P0 and from 0.5 ± 0.6 Pa to 54.3 ± 7.6 Pa (n = 3) for P50 (Figure 3b).

The Plateau modulus G_0_ changed from 6.4 ± 0.7 Pa to 742.5 ± 59.9 Pa (n = 3) for P0 and from 16.8 ± 0.323 Pa to 322.1 ± 14.5 Pa (n = 3) for P50 (Figure 3c).

The static viscosity η_0_ increased from 0.083 × 10^6^ ± 0.009 × 10^6^ Pa·s to 5.092 × 10^6^ ± 0.196 × 10^6^ Pa·s (n = 3) for P0 samples, and from 0.130 × 10^6^ ± 0.036 × 10^6^ to 1.556 × 10^6^ ± 0.075 × 10^6^ Pa·s (n = 3) for P50 samples, similar to the increase in concentration (Figure 3d).

In order to investigate the differences in the structural and functional properties of hydrogels with similar viscoelastic properties, but made from pectins characterized by different DEs, the following two pairs of hydrogels were taken for extended reciprocal characterization: P0-1.25% and P50-3%, P0-1.5% and P50-4.5%, respectively. Similar rheological properties in these pairs were achieved by adjusting the concentration of pectin according to the principle that an increase in DE requires an increase in the concentration of pectin due to a decrease in the number of free carboxyl groups, resulting in a low cross-link density.

### 2.4. Scanning Electron Microscopy

Mesoporous structures were obtained by the critical point drying of pectin-based hydrogels (Figure 4a). The morphology and architecture of micro- and nanopores are typical of lyophilized sponges [15].

SEM micrographs of pectin hydrogel sponges are presented in Figure 5. These images demonstrate the intricate and interconnected network of mesopores, whose size are reduced and density increased by the growth concentration of pectin. Pore size of the samples P50-4.5% (Figure 5a) and P50-3% (Figure 5c) was 88 ± 5 nm (n = 100), which was significantly different from the sample P50-1.5% (Figure 5e); both P0-1.5% (Figure 5b) and P0-1.25% (Figure 5d) showed a bigger pore size of 115 ± 4 nm (n = 100). The difference between the pore size of a pair of sponges prepared from hydrogels with the same G′ (P0-1.5% and P50-4.5%, P0-1.25% and P50-3%) is clear and shown in Figure 5f.

The thickness of fibrils increased with the increasing concentration of gels. In this case, P0 hydrogels were characterized by a rapid thickening of fibrils with a relatively small change in concentration, while for P50 gels, an increase in concentration from 1.5% to 4.5% did not significantly affect the thickness of the fibrils. For P0-1.25%, the fibril thickness was 19 ± 5 nm (n = 100); for P0-1.5%, it was 43 ± 9 nm (n = 100). P50 gels showed a change in the average fibril thickness in a narrow range from 31 ± 5 to 35 ± 6 nm (n = 100), which is shown in Figure 5g.

### 2.5. Gel Fraction

The gel fraction of hydrogels with different contents of DE and pectin is shown in Figure 6a. The gel fraction increased with an increase in the number of methoxylated groups in the pectin sample. No dependence of the gel fraction on the pectin content in the sample was found. The gel fraction of gels P50-4.5% and P0-1.5% with the high meanings of G′ were not significantly different, in contrast to pairs P50-3% and P0-1.25% whose gel fractions were different (*p* < 0.05).

### 2.6. Swelling Test and In Vitro Stability

Simulated body fluid (SBF) was used to evaluate the swelling properties of hydrogels. The swelling profiles as a function of time are shown in Figure 6b, and the corresponding maximum swelling ratios (MSR) after reaching equilibrium are shown in Figure 6c. The test shows that pectin-based hydrogels with DE 0% swelled completely in the first 10 min of incubation in SBF; while for hydrogels with DE 50%, this took up to 30 min at a concentration by weight of 3% and 1.5% and up to 4 h for the same hydrogel at a concentration of 4.5% (Figure 4b). In P50 samples, the time of achieving a state of equilibrium was increased with the rise of pectin concentration by weight, but meanings of MSRs were similar. MSRs of P0-1.5% and P50 were in one range from 618 ± 71% to 744 ± 97% (n = 5), while the MSR of P0-1.25% was 930 ± 66%.

### 2.7. Water Holding Capacity

The water retention capacity of hydrogels is shown in Figure 6d. The water holding ratios for all gels decreased rapidly for less than a day until reaching zero values. The complete drying of hydrogels was observed for up to 18 h. The P50 samples retained moisture more efficiently than the P0 samples. For P50, the total drying time was 16–18 h, and for P0 it was only 14 h. There was a slight difference between P0-1.25% and P0-1.5% at the beginning of the experiment; the maximum swelling coefficient of P0-1.25% was greater than P0-1.5%. It affected the moisture retention capacity during almost the entire experiment; however, after 10 h, the difference was leveled.

### 2.8. Cell Behavior Analysis

Cell behavior analysis was performed after 5 days of cultivation. Cell viability, measured by trypan blue staining, did not differ significantly when cultured on all hydrogels compared to control cells cultured on polystyrene and ranged from 90.47 ± 3.04% to 95.85 ± 1.84% (n = 3) for C6 and from 93.45 ± 4.08% to 96.86 ± 1.77% (n = 3) for U87-MG, depending on the hydrogel type (Figure 7a).

The evaluation of metabolic activity was carried out using an MTT assay. It was shown that hydrogels significantly retarded the metabolism of tumor nerve cells. At the same time, pectin-based hydrogels with DE 50% had a stronger inhibitory effect than pectin-based hydrogels with DE 0%, which had similar rheological properties. For example, when cells were cultivated on P50-4.5% hydrogel, their metabolic activity decreased by 3.3-fold for C6 and 1.57-fold for U87-MG, compared to P0-1.5%, and P50-3% compared to P0-1.25% reduced metabolic activity by 4.53-fold and 2.11-fold for C6 and U87-MG, respectively (n = 8) (Figure 7b).

The representative image series captured by the Cell-iQ high-content microscopy analyser for migrating U87-MG cells cultured on different substrates is shown in Figure 8. The quantitative analysis of migration activity showed a cell-type dependent effect (Figure 7c). The migration rate of C6 cells on all hydrogels decreased and was 6.69 ± 0.051 µm/h on polystyrene and from 3.65 ± 0.43 to 5.60 ± 0.49 µm/h on carbohydrate hydrogels, depending on the type of material. The migration rate of U87-MG cells was significantly lower (about 15%) than on polystyrene and was 6.93 ± 0.22 µm/h on polystyrene and from 5.67 ± 0.40 to 6.31 ± 0.27 µm/h on pectin hydrogels, depending on the type of material. Pectin-based hydrogels with DE 50% slowed down the migration of C6 cells significantly more than pectin-based hydrogels with DE 0% with similar rheological properties. P50-4.5% reduced the cell migration rate by 30% compared to P0-1.5% and by 56% compared to polystyrene. P50-3% reduced the cell migration rate by 28% compared to P0-1.25% and by 56% compared to polystyrene.

## 3. Discussion

We demonstrated that a set of biocompatible materials mimicking the native ECM matrix of the brain can be obtained by the ionic gelation of pectins with different DEs. The comprehensive physicochemical characterization of the created materials is necessary for fine-tuning the properties, which is important for applications such as tissue engineering and the creation of bioactive matrices, in which the mechanical microenvironment controls the cell fate [16,17]. Recent studies proposed several successful protocols for the development of pectin-based hydrogels suitable for prospective anti-cancer therapy. They focused on hydrogels produced by cross-linking with aldehyde derivatives formed via the oxidation of hydroxyl groups. These hydrogels were also supplemented by a natural or chemically derived co-polymer to reach the optimal mechanical properties of hydrogels and to ensure drug immobilisation. An injectable, self-healing hydrogel system from oxidized pectin was made in combination with chitosan and γ-Fe_2_O_3_supplements [18]. Pectin-based biodegradable self-healing hydrogels were produced by cross-linking polymer chains using pectin aldehyde (pectin-CHO) and acylhydrazide functionalized polymer poly(N-isopropylacrylamide- stat -acylhydrazide) P(NIPAM- stat -AH) [19]. Pectins may serve as valuable supplements for chitosan-based hydrogel formation, improving the mechanical properties of the biomaterials developed [20]. In this study, we focused on the elaboration of the hydrogels based on pectin alone, without any polymeric additives, exploiting ionic gelation in physiological conditions. This became possible by tuning the viscoelastic properties and the porosity of pectin hydrogels via regulating the number of free carboxyl groups in combination with the adjustment of pectin concentration. We presented data evaluating the mechanical and structural properties of pectin hydrogels with particular implications for the development of biomaterials that could be promising for anti-glioma therapy.

The formation of a three-dimensional mesh of hydrogel structure was predicted by rheological measurements and confirmed by SEM. Rheological analysis providing the determination of storage (G′), loss (G″), plateau (G_0_) moduli, and static viscosity (ƞ_0_) showed that the properties of pectin-based hydrogels with different DEs varied significantly depending on the pectin content in the sample. Hydrogel formation was proven for each sample by checking the correspondence G′ > G″, and the substantial growth in the storage modulus was concentration-dependent.

All of the samples were classified as strong gels, for which G′ was at least 10 times higher than G″ [21]. This may be useful, given that in the ECM of the CNS, elastic behavior prevails over viscous behavior [22], and the elastic modulus varies for different regions from 100 to 1000 Pa, which corresponds to the mechanics of pectin hydrogels. In addition, it was shown that neurons are able to perceive the rigidity of the environment in vitro and prefer soft substrates [23,24]. At the same time, the stiffness of a substrate was shown to affect glioma cell migration and proliferation, specifically glioma cells spread more extensively and rapidly on hard ECM and glioma cell migration decreased with the decreasing rigidity of ECM [25].

For comparative analysis, we selected pairs of pectin-based hydrogels with different DEs but with similar storage and loss moduli. Despite the similar mechanical behavior, the morphology of the gels changed in dependence of DE.

The mesoporous structure of the hydrogels, which is quite easy to control by adjusting the pectin concentration together with the appropriate DE, as shown in Figure 5, made them convenient for loading active ingredients or cells. At the same time, the ability to modulate high values of G′ made them more resistant to mechanical influences in vivo.

The compared pairs of samples (P0-1.5% vs. P50-4.5%, and P0-1.25% vs. P50-3%) significantly differed both in the morphology of the hydrogels, shown by SEM, and in the swelling properties. Hydrogels based on fully de-esterified pectin (DE 0%) had larger pores due to the low polymer concentration required to achieve the desired stiffness, which made the pore structure much more disordered, leading to the easy penetration of water molecules. In addition, the saturation of pectin chains with carboxyl groups led to an increase in the formation of hydrogen bonds, resulting in a better interaction of the polymer with water molecules. The high content of free carboxyl groups increased the electrostatic repulsion in the hydrogel, which favoured the expansion of the hydrogel network [26]. Apart from this, the osmotic pressure caused by enrichment with carboxyl groups also increased the water absorption capacity [27].

The difference in physicochemical and mechanical properties affected the biological properties of materials. While maintaining high cell viability by all the materials, the pectin-based matrix with DE 50% had a greater repressive effect on the metabolic activity than the materials with DE 0, determined by the MTT assay, which usually indicates a decrease in cell proliferation. The cell migration velocity of brain tumor cells on all pectin hydrogels reduced by 28–56% compared to the control series. Hydrogels with smaller pores (P50-3% and P50-4.5%) significantly reduced the cell migration velocity compared to materials with an equivalent G′ but larger pores (P0-1.25% and P0-1.5%, respectively).

The development of bioactive materials for the treatment of brain tumors is one of the key tasks of modern biomedical research and tissue engineering. Traditional approaches to anti-cancer therapy are not effective enough and are often unable to increase the long-term survival of patients [28]. A noticeable disadvantage of traditional therapeutic strategies of glioblastoma is the one-sided focus on the destruction of glioblastoma cells and the lack of consideration of factors affecting the microenvironment. An altered ECM is currently thought to be a key factor in tumor development and progression, but the control of this aspect of the tumor microenvironment has yet to be exploited for clinical application in the treatment of glioblastoma [3]. Carbohydrate-based biomaterials can be used for the target therapy of glioblastoma, since carbohydrates are widely present in the ECM of the brain, and the use of radiotherapy aggravates pathogenesis with enhanced fibrosis and an increase in matrix rigidity due to the production of protein components [29,30]. Pectin-based biomaterials mimic the native ECM of the brain and help reduce the aggressiveness and invasiveness of tumor cells by enriching the tumor microenvironment with carbohydrates that can reduce the proliferation and migration ability of cancer cells.

Biomaterials can promote cell recruitment to the implantation site after damage to the nervous system [31,32]. The anti-tumor biomaterial should support the migration of resident cells from the edge of the resection cavity after tumor resection to a lesser extent than the tumor ECM. At the same time, the properties of the implanted material should surpass the malignant-enhancing properties of hyaluronic acid in the brain, either by disrupting binding or by providing a slowdown in the metabolism of contacting cells. Pectin-based materials, unlike hyaluronic acid, reduce the migration rate and provide an effective slowdown in the metabolism of tumor cells.

A number of authors suggest using biomaterials based on low molecular weight hyaluronic acid instead of high molecular weight, which promotes local anti-tumor inflammation and affects growth factor signaling [33,34]. Previous experiments with low-molecular-weight pectins showed significant anti-proliferative and anti-migration effects on several cancer cells [35,36]. In addition to anti-tumor effects, chemotherapeutic biomaterials can also be designed with pro-regenerative signals to stimulate the growth of healthy brain parenchyma as tumor tissue is eliminated [37]. We demonstrated that hydrogels based on modified pectins with controlled viscoelastic properties significantly reduce the metabolic activity of glioblastoma cells without exerting a toxic effect on them and can retard the migration of cancer cells.

In the present study, hydrogels based on pectins with various degrees of esterification were developed, and their structural properties, rheological properties, and swelling behavior were characterized. Materials showed tunable mechanical properties, with Young’s modulus values comparable to the ECM stiffness of brain tissue. The developed hydrogel formulations showed a variable stiffness in relation to different DEs and the final concentration of the polymer. We demonstrated the ability of pectin-based hydrogels to dramatically reduce the metabolic activity of tumor cells upon contact, while not causing cell death and maintaining high cell viability for a long time. Their effect on migration activity turned out to be cell-specific and reduced the cell migration velocity. Thus, polysaccharide hydrogels based on modified pectins could be promising for combined therapy against glioma.

## 4. Materials and Methods

### 4.1. Preparation of Modified Pectin

Preparations of modified pectins with an esterification degree close to 0% and 50% were obtained from highly esterified citrus pectin with an esterification degree of 60.2% (Copenhagen Pectin A/S, Lille Skensved) by alkaline de-esterification, as was previously described by Khotimchenko and colleagues [11]. Highly esterified citrus pectin was washed with ethanol and dried at 70 °C. After drying, the pectin sample was ground and fractionated according to the particle size. In the study, the fine fraction was sieved through a sieve with a cell size of 74 μm. The process of alkaline de-esterification of pectin performed at pH > 8.5 was preceded by the initial neutralization of free carboxyl groups of anhydrogalacturonic acid. When the desired DE was achieved, the reaction mixture was acidified with 1 M HCl solution in 50% ethanol, reaching a pH of 5–6 with vigorous stirring. The obtained pectin preparation was separated from the water–ethanol solution by filtration and first washed with 300 mL of a 50% ethanol solution and then with 150 mL of 95% ethanol. Washed pectin was dried at 70 °C. This method allowed us to obtain a set of products that differed by the content of methoxylated groups. Pectin sample characterization was performed according to the previously described technique.

### 4.2. Hydrogel Preparation

Modified pectin powders were dissolved in deionized water, and the concentration of the powder was from 0.5% to 5% by weight. Samples were allowed to soak for 1 h at room temperature for hydration. Polysaccharides were dissolved in a water bath for 1 h at 95 °C for pectin with DE 0% (P0) and for 15 min at 60 °C for pectins with DE 50% (P50) with constant stirring. The resulting colloidal solutions were clarified by centrifugation at 3000× *g* for 40 min. Clarified colloidal solutions were transferred into new polypropylene tubes and sterilized by autoclaving at 105 °C, 0.7 atmospheres, 10 min.

Gels were formed by ionic gelation by binding the carboxyl groups of pectin with Ca^2+^ ions contained in solutions of gelation initiators. We used sterile gel initiator solutions containing 100 mM N-(2-hydroxyethyl)-piperazine-N′-2-ethanesulfonic acid (HEPES) pH 7.4, 300 mM NaCl, and CaCl_2_ in an amount sufficient to complete gelation (Table 2). Pectin-based hydrogels were prepared by equal mixing volumes of solutions of pectins with gelation initiators at 4 °C. They were incubated for 30 min for complete gelation.

### 4.3. Morphological Analysis

The morphology of the hydrogels was characterized by a field emission scanning electron microscope (SEM) Sigma 300VP (Carl Zeiss, Jena, Germany). The samples were first placed sequentially in DMEM, PBS, 20%, 40%, 60%, 80%, and 100% acetone for 10 min and treated in a critical point dryer Leica EM CPD030 (Leica Microsystems, Wetzlar, Germany). Small pieces of the dried samples were placed on copper objective tables covered by electroconductive tape. Upper parts of the samples were cut. Freshly prepared surfaces were coated with a nanometer-thick chromium layer. Observations were carried out at the accelerating voltage of 10 kV and vacuum of 10-5 Torr. The pore size was analyzed by stochastically measuring the aperture of three different fields of vision using Zeiss SmartTiff software. The diameter was measured as an average of two measurements for each pore, 100 pores in total per sample.

SEM studies were performed with the support of Far Eastern Center for Electron Microscopy located in A.V. Zhirmunsky National Scientific Center of Marine Biology FEB RAS (Vladivostok, Russia).

### 4.4. Rheological Analysis

Rheological properties of pectin hydrogels were performed using a HAAKE Mars-III rheometer (Thermo Fisher Scientific Inc., Waltham, MA, USA) equipped with a 35 mm serrated parallel plate geometry. Measurements were produced in frequency dependence with constant shear stress and creep/recovery modes. Immediately following the preparation of hydrogel samples, G′ and G″ were measured at a constant shear stress of τ_0_ = 1 Pa and frequency of 1 Hz (from 100.0 Hz to 0.001 Hz log during measuring). The solutions were kept and mixed at 4 °C, and gelation and measurements were performed at 37 °C. Each test was performed in triplicate. Additionally, the plateau modulus G_0_ and static viscosity η_0_ were measured when applying and removing a constant shear stress τ_0_ at the creep mode (Table 3). Both creep and recovery stages were 1800 s at 37 °C.

### 4.5. Fourier Transform Infrared (FTIR) Spectra

FTIR spectra were obtained to understand the molecular interactions and functional group characterization. The FTIR spectra were obtained using an IRAffinity-1S spectrophotometer (Shimadzu, Kyoto, Japan) in ATR-FTIR mode. All spectra were recorded with the resolution of 4 cm^−1^ in the range of 400–4000 cm^−1^ with 20 scans.

### 4.6. Gel Fraction

The fresh hydrogels were dried to a constant weight, then immersed in simulated body fluid (SBF) [38] at room temperature until the non-crosslinked parts were completely removed, followed by lyophilization and recording the final weight. The gel fraction was calculated as follows [39]:(1)$$\mathrm{Gel}\;\mathrm{fraction}\;(\%)=\frac{W_{e}}{W_{0}},\;\times \;100\%$$
where W_0_ and W_e_ represent the weight of dried hydrogels before and after extraction. All the measurements were repeated 6 times, and the mean value was presented with the standard deviation as the error bar.

### 4.7. Water Holding Capacity

The water holding capacity was tested according to Qing et al. [39]. The samples were immersed in SBF to reach swelling equilibrium. After wiping off the surface water, their weights were recorded and denoted as W_d_. Then, the wet hydrogels were kept at 37 °C in the thermostat. Their weights were measured and recorded at predetermined time intervals. The water retention ratio of the hydrogels was defined as follows:(2)$$\mathrm{Water}\;\mathrm{Holding}\;\mathrm{Ratio}\;(\%)=\frac{W_{d}}{W_{0}},\;\times \;100\%$$
where W_d_ and W_0_ represent the time-dependent and initial (t = 0 min) weight of hydrogels.

### 4.8. Swelling Behavior

The swelling behavior of hydrogels was evaluated in simulated body fluid (SBF) (pH = 7.4) [38]. The dried samples were immersed in SBF and taken out at regular time intervals for weight measurement. The residual liquid on the hydrogel surface was wiped quickly with filter paper. The sample weight was subsequently measured and recorded. The swelling ratio (SR) of the hydrogels was determined as follows:(3)$$\mathrm{Swelling}\;\mathrm{Ratio}\;(\%)=\frac{W_{t}-W_{0}}{W_{0}},\;\times \;100\%$$
where W_0_ and W_t_ denote the weight of dried and swelling hydrogels at regular time intervals. When the swelling reached equilibrium, the corresponding swelling ratio was taken as the maximum swelling ratio (MSR).

### 4.9. Cell Cultures

#### 4.9.1. Cell Lines

Rat glioma C6 cells and human malignant glioma U87-MG cells were obtained from ATCC (Manassas, VA, USA) and cultured in DMEM/Ham’s F-12 media, with L-Glutamine (Capricorn, Bergkamen, Germany) supplemented with 10% fetal bovine serum (FBS; Gibco, Shanghai, China), 100 U/mL penicillin, and 100 μg/mL streptomycin, at 37 °C in a humidified atmosphere of 5% CO_2_. After cells reached 70% confluence, the cells were harvested, washed with a PBS buffer, resuspended, and used for experiments.

#### 4.9.2. Cell Viability and Metabolic Activity Analysis

Cells were seeded in a 96 well culture plate (Greiner) at a density of 2 × 10^3^ cells/well on hydrogels or polystyrene in full DMEM/F12 media for 5 days at 37 °C, 5% CO_2_. After treatment, viability was measured by trypan blue staining. The metabolic activity analysis was performed using MTT tetrazolium dye. MTT was added to each well at a final concentration of 0.5 mg/mL, and the cells were further incubated at 37 °C for 4 h. Lysis buffer (isopropanol solution with 1% NP-40, 4 mM HCl) 150 μL was added to each well after removing the medium. After resuspension and 30 min incubation, the plate was centrifuged at 1000× *g* for 10 min. The supernatant was transferred to new wells, and the absorbance was measured at 595 nm using a microplate absorbance reader (iMark, Bio-Rad, Hercules, CA, USA).

#### 4.9.3. Evaluation of Cell Migration on Hydrogels

Cells were seeded in a 24 well culture plate (Greiner) at a density of 1.5 × 10^4^ cells/well on hydrogels or polystyrene in full DMEM/F12 media and incubated for 5 days at 37 °C, 5% CO_2_ in automated cell culture and the analysis system Cell-IQ^®^ (CM Technologies Oy, Tampere, Finland). Cells were visualized by phase contrast using Nikon Plan Fluor 10× objective (NA 0.30). Each visual field was visualized for 4 h. Cell migration activity was measured using the Cell-IQ Analyzer software.

### 4.10. Data Analysis

All experiments, unless differently specified, were performed in triplicate, and quantitative data were shown as mean ± standard deviation after an analysis of variance (ANOVA), which enabled comparison of the data from the study groups via the GraphPad Prism software (GraphPad Software Inc., San Diego, CA, USA). Statistical differences were designated significant if the *p*-values were less than 0.05 (* *p* ≤ 0.05) and highly significant if the *p*-values were less than 0.01 (** *p* ≤ 0.01), less than 0.001 (*** *p* ≤ 0.001), or less than 0.0001 (**** *p* ≤ 0.0001).

The Mann-Whitney U-test was used to assess the differences between two independent samples.

## 5. Conclusions

By applying the data presented, it is possible to conduct future in vitro and in vivo experiments to determine the optimal ratio of concentration and DE of pectin, which is best suited for therapy against glioma by controlling the rate of migration and proliferation of tumor cells, while maintaining the possibility of neurite outgrowth. Carbohydrate-based hydrogels are promising for use in the treatment of glioblastoma, because polysaccharides and proteoglycans are the key components in neural ECM, supporting the structure of the brain and controlling cell fate. At the same time, unlike hyaluronic acid, which is most widely represented in the ECM of the brain, plant polysaccharides, such as pectins, do not accelerate the proliferation and migration of glioblastoma cells. By changing the percentage of free carboxyl groups and the concentration of pectin, the viscoelastic properties, porosity, water-retaining capacity of hydrogels, and the ability to immobilize various substances can be easily adjusted, which allows for the creation of hydrogels with precisely matched characteristics, which is valuable for anti-glioma therapy.

## Figures and Tables

**Figure 1 ijms-24-00630-f001:**
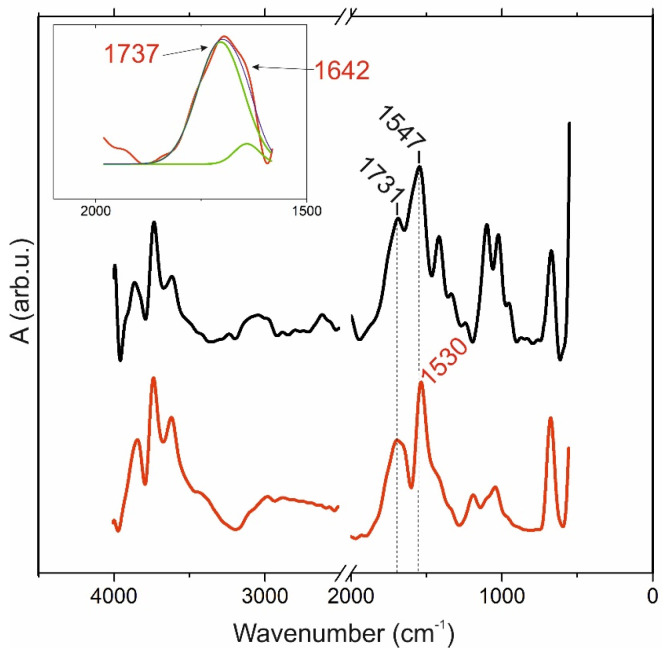
FTIR absorption spectra: black spectrum, powder of pectin with DE 0%; red spectrum, gel obtained from pectin 0 solution with a concentration of 1.25%. The selected area is the range of vibrations of carboxyl groups from the spectrum of the gel.

**Figure 2 ijms-24-00630-f002:**
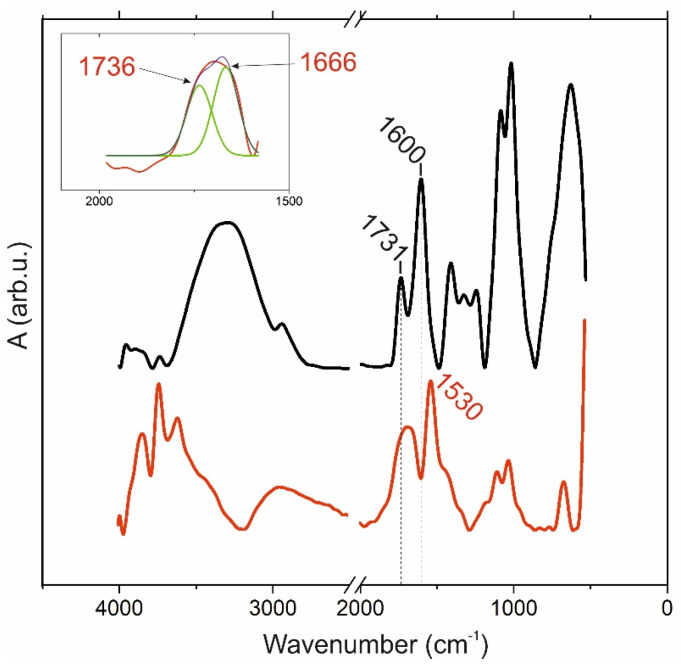
FTIR absorption spectra: black spectrum, powder of pectin with DE 50%; red spectrum, gel obtained from pectin 50 solution with a concentration of 3%. The selected area is the range of vibrations of carboxyl groups from the spectrum of the gel.

**Figure 3 ijms-24-00630-f003:**
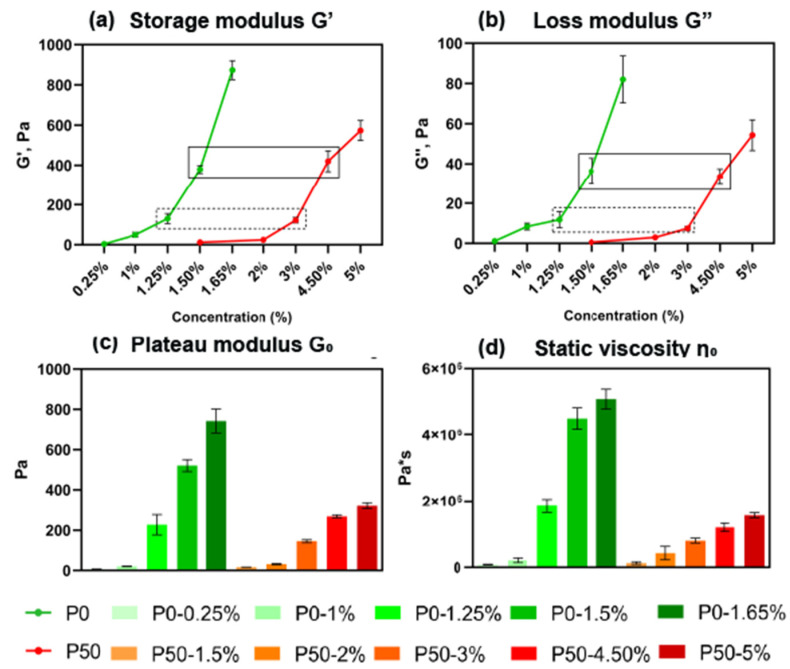
Rheological properties of pectin hydrogels: (**a**) storage modulus G′; (**b**) loss modulus G″; (**c**) plateau modulus G_0_; (**d**) static viscosity η_0_, n = 3.

**Figure 4 ijms-24-00630-f004:**
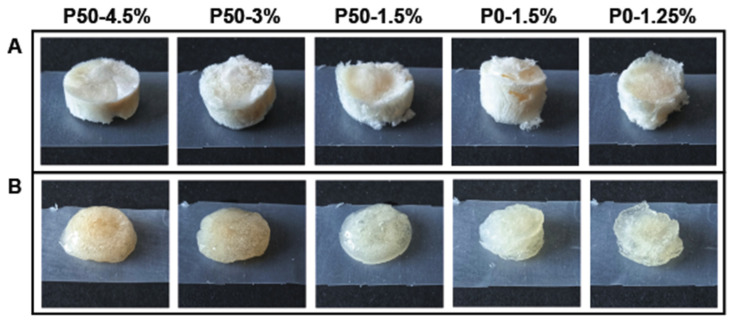
Image of pectin-based materials: (**A**) pectin freeze-dried sponges; (**B**) completely swollen hydrogels.

**Figure 5 ijms-24-00630-f005:**
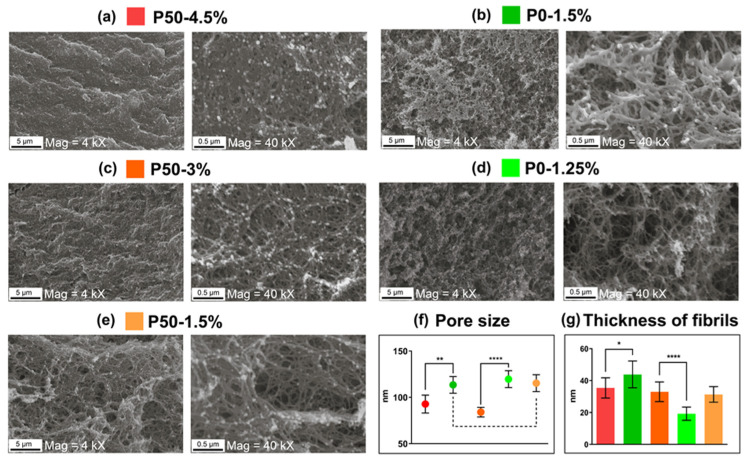
SEM of pectin hydrogels at magnifications of 4 kX and 40 kX: (**a**) P50-4.5%; (**b**) P50-3%; (**c**) P50-1.5%; (**d**) P0-1.5%; (**e**) P0-1.25%; (**f**) pore size; (**g**) fibril thickness, n = 100, * *p* < 0.05, ** *p* < 0.01, **** *p* < 0.0001.

**Figure 6 ijms-24-00630-f006:**
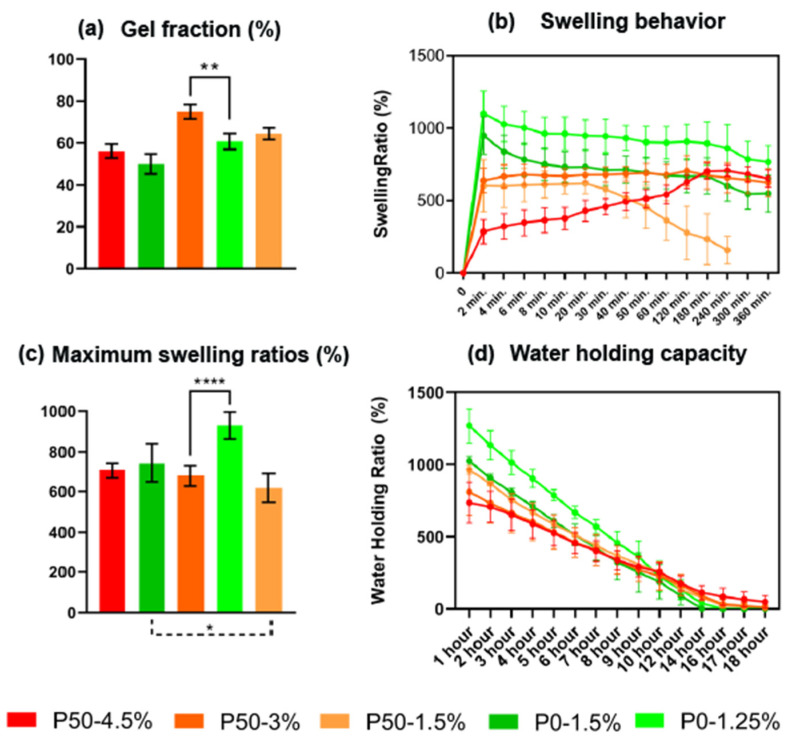
(**a**) Gel fraction; (**b**) swelling behavior; (**c**) maximum swelling ratios; (**d**) water holding capacity of pectin-based hydrogels, n = 5, * *p* < 0.05, ** *p* < 0.01, **** *p* < 0.0001.

**Figure 7 ijms-24-00630-f007:**
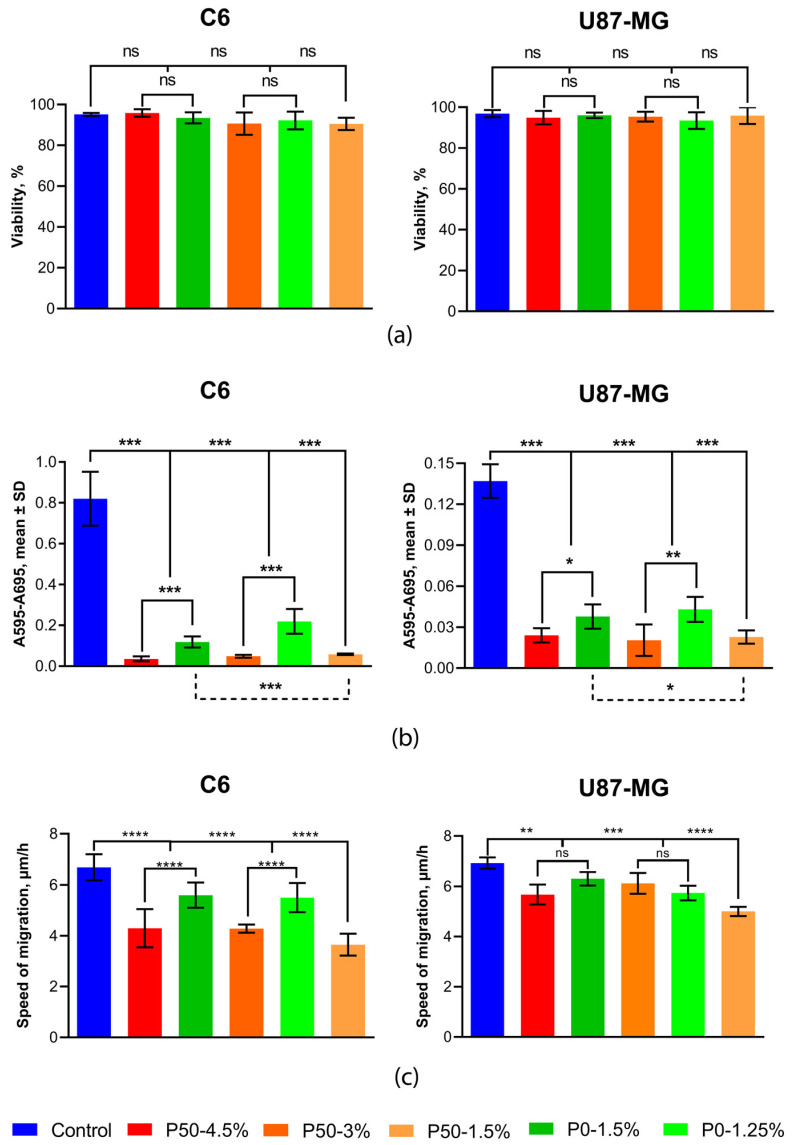
The effect of pectin hydrogels on tumor cells: (**a**) cell viability, (**b**) metabolic activity, (**c**) migration activity, * *p* < 0.05, ** *p* < 0.01, *** *p* < 0.001 **** *p* < 0.0001.

**Figure 8 ijms-24-00630-f008:**
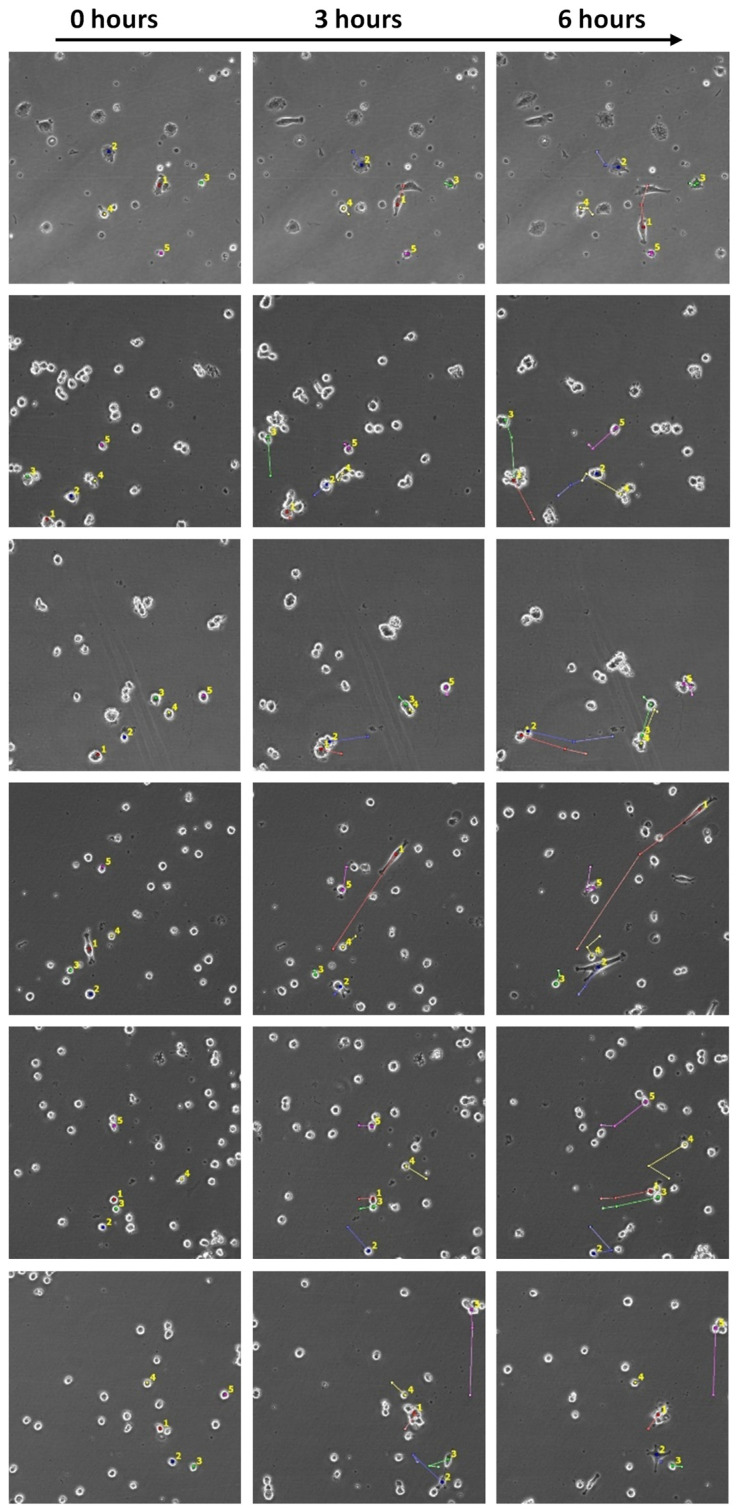
Representative images of migrating cells. Colored lines correspond to the cell track. Phase contrast microscopy, 10× magnification.

**Table 1 ijms-24-00630-t001:** Physicochemical parameters of pectin preparations.

Parameter	P0	P50	Original Pectin
Degree of esterification, %	1.2	52.0	60.2
Total anhydrogalacturonic acid content, %	73.7	72.0	79.4
Non-methylated anhydrogalacturonic acid content, %	72.8	34.6	31.6
Intrinsic viscosity, mL/g anhydrogalacturonic acid	408	570	915

**Table 2 ijms-24-00630-t002:** Ca^2+^ content in the gel initiator solution.

Sample	Ca^2+^ Content (mM)	Sample	Ca^2+^ Content (mM)
P50-0.5%	4	P0-0.25%	0.666
P50-1%	8	P0-0.5%	1.333
P50-1.5%	12	P0-1%	2.666
P50-2%	16	P0-1.25%	3.333
P50-3%	24	P0-1.5%	4
P50-4%	32	P0-1.65%	4.4
P50-4.5%	36	P0-3%	8
P50-5%	40		

**Table 3 ijms-24-00630-t003:** Constant shear stress parameter for creep and recovery mode.

Sample	Shear Stress	Sample	Shear Stress
P50-1.5%	2.5	P0-0.25%	1.5
P50-2%	2	P0-1%	2
P50-3%	3.5	P0-1.25%	3
P50-4.5%	4	P0-1.5%	4
P50-5%	5	P0-1.65%	3.5

## Data Availability

Images and data are available from the corresponding author upon reasonable request.

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
