# Peer review of "Preparation of Hydrogels Based on Modified Pectins by Tuning Their Properties for Anti-Glioma Therapy"

_ijms, 2022, doi:10.3390/ijms24010630_

Round 1

Reviewer 1 Report

The authors have characterized pectin based hydrogels for its use an anti glioblastoma treatment. 

Althought it is an interesting work, it needs further changes. First of all I suggest to include in the introduction section more information regarding the vision of the work. Maybe it would be good to include an explanation about the current treatment and the difficulties.

Regarding the results section:

Please, check the figure number in the line 120 and add a reference in line 124.

Figure 3 seems to have the order changed: please check the titles of the graphs and verify them in the text. Moreover, the legend of the figure 3 has an error: P5 (check and correct it to P50)

Can you explain better the paragraph between lines 166-168?

In section 2.5 it would be better if you add the information regarding the figure number before.

The graphs titles do not correspond to the figure legend in figure 6. Please, check them also in the text.

The authors could change the order of the graphs in figure 7. As they first mention the cell viability, I would add first this information, after the MTT assay and at the end the migration analysis. Moreover, regarding the migration analysis I would suggest to rewrite the last paragraph (lines 237-247).

Regarding the discussion section I would suggest its improvement. In general there is little comparison with other studies. Are pectin based hydrogels being used for other type of cancers? There are different reports analyzing chitosan/pectin based hydrogels, what are the advantages of the system they have proposed?

To finish, I would suggest the authors to improve the figure quality and to check the styling of the manuscript (for instance, there are some references in bold)

Author Response

Dear reviewer, we greatly appreciate the time and effort You invested in our article. Your notes have greatly improved our work and we are very thankful for Your feedback. Your recommendations helped us to better emphasize new findings in this field. We have created an additional image illustrating cell migration on different substrates. The text of the manuscript and figures have been revised. Information about current problems and promising approaches in the treatment of glioblastoma has been added to the introduction and discussion.

Point 1: Althought it is an interesting work, it needs further changes. First of all I suggest to include in the introduction section more information regarding the vision of the work. Maybe it would be good to include an explanation about the current treatment and the difficulties.

Response 1: Introduction section has been improved. Information about the disadvantages of the current treatment of glioblastoma has been added (line 45-50).

Point 2: Please, check the figure number in the line 120 and add a reference in line 124.

Response 2: Description of Figures 1-2 and references have been checked and corrected.

Point 3: Figure 3 seems to have the order changed: please check the titles of the graphs and verify them in the text. Moreover, the legend of the figure 3 has an error: P5 (check and correct it to P50).

Response 3: Figure 3 and section 2.3. has been revised.

Point 4: Can you explain better the paragraph between lines 166-168?

Response 4: The paragraph has been rewritten to better explain the choice of sample pairs selected for the study (lines 182-189).

Point 5: In section 2.5 it would be better if you add the information regarding the figure number before.

Point 6: The graphs titles do not correspond to the figure legend in figure 6. Please, check them also in the text.

Response 5-6: Section 2.5 and Figure 6 has been revised.

Point 7: The authors could change the order of the graphs in figure 7. As they first mention the cell viability, I would add first this information, after the MTT assay and at the end the migration analysis. Moreover, regarding the migration analysis I would suggest to rewrite the last paragraph (lines 237-247).

Response 7: Figure 7 has been corrected. Figure 8 demonstrating cell migration tracks on different materials has been added (line 278).

Point 8: Regarding the discussion section I would suggest its improvement. In general there is little comparison with other studies. Are pectin based hydrogels being used for other type of cancers? There are different reports analyzing chitosan/pectin based hydrogels, what are the advantages of the system they have proposed?

Response 8: A paragraph devoted to comparing the materials we created with other studies was added to the discussion (lines 287-302).

Point 9: To finish, I would suggest the authors to improve the figure quality and to check the styling of the manuscript (for instance, there are some references in bold).

Response 9: Done.

Reviewer 2 Report

The study Belousov et al. presents an innovative method of preparation of hydrogels, based on modified pectin properties, for anti-glioma therapy. The results are potentially applicable for postoperative therapy for gliomas to reduce migration and proliferation of tumor cells and to support the healthy microenvironment. The manuscript is well-designed and is of significant interest to readers but needs improvement in presentation. I recommend the manuscript for publication with thew following suggestions:

1.  Please include a specific section “Abbreviations” with all abbreviations used in the manuscript.

2.  It will be nice include examples of images of cell migration on different types of hydrogel in the supplemental file or in the figure. 

3.  Please include the number of experiments (n) after each presented average (not just in the method section).

4.  Line 17: to 0 and 50% change to 0% and 50%

5.  Line 40: include space between method and [5,6].

6.  Line 99: DE 50?

7.  Fig 4, line 174: Change (a) and (b) to (A) and (B) 

Author Response

Dear reviewer, thank you very much for Your feedback and recommendations on our work. Your notes allowed us to improve the article.

Point 1: Please include a specific section “Abbreviations” with all abbreviations used in the manuscript.

Response 1. “Abbreviations” section has been added to the manuscript.

Point 2: It will be nice include examples of images of cell migration on different types of hydrogel in the supplemental file or in the figure. 

Response 2. Example images of migrating cells have been added to the manuscript as Figure 8.

Point 3: Please include the number of experiments (n) after each presented average (not just in the method section).

Response 3. Done.

Point 4: Line 17: to 0 and 50% change to 0% and 50%.

Response 4. Done.

Point 5: Line 40: include space between method and [5,6].

Response 5. Done.

Point 6:Line 99: DE 50?

Response 6: DE 50 changed to DE 50%.

Point 7: Fig 4, line 174: Change (a) and (b) to (A) and (B).

Response 7. Done.

Round 2

Reviewer 1 Report

The authors have modified and improved enough the manuscript for its publication